# Influence of Carrier Filling Ratio on the Advanced Nitrogen Removal from Wastewater Treatment Plant Effluent by Denitrifying MBBR

**DOI:** 10.3390/ijerph16183244

**Published:** 2019-09-04

**Authors:** Yuanzhe Zhao, Quan Yuan, Zan He, Haiyan Wang, Guokai Yan, Yang Chang, Zhaosheng Chu, Yu Ling, Huan Wang

**Affiliations:** 1State Key Laboratory of Environmental Criteria and Risk Assessment, Chinese Research Academy of Environmental Sciences, Beijing 100012, China; 2Engineering Center for Environmental Pollution Control, Chinese Research Academy of Environmental Sciences, Beijing 100012, China; 3Beijing China’s Sustainable Development Water Purification Material Technology Co., Beijing 100012, China; 4National Engineering Laboratory for Lake Pollution Control and Ecological Restoration, Chinese Research Academy of Environmental Sciences, Beijing 100012, China

**Keywords:** denitrifying MBBR, polyethylene carrier, filling ratio, T-RFLP, qPCR

## Abstract

The filling ratio (FR) of a carrier has an influence on the pollutant removal of the aerobic moving bed biofilm reactor (MBBR). However, the effect of the polyethylene (PE) carrier FR on the performance and microbial characteristics of the denitrifying MBBR for the treatment of wastewater treatment plant (WWTP) effluent has not been extensively studied. A bench-scale denitrifying MBBR was set up and operated with PE carrier FRs of 20%, 30%, 40%, and 50% for the degradation of chemical oxygen demand (COD) and nitrogen from WWTP effluent at 12 h hydraulic retention time (HRT). The nitrate removal rates with FRs of 20%, 30%, 40%, and 50% were 94.3 ± 3.9%, 87.7 ± 7.3%, 89.7 ± 11.6%, and 94.6 ± 4.0%, and the corresponding denitrification rates (r_NO3--N_) were 8.0 ± 5.6, 11.3 ± 4.6, 11.6 ± 4.6, and 10.0 ± 4.9 mg NO_3_^−^-N/L/d, respectively. *Nitrous oxide reductase* (*nosZ*) gene-based terminal restriction fragment length polymorphism (T-RFLP) analysis illustrated that the highest functional diversity (Shannon’s diversity index, H′) of biofilm microbial community was obtained at 30% FR. The quantitative polymerase chain reaction (qPCR) results indicated that the abundance of *nitrate reductase* (*narG*) and *nosZ* genes at 30% FR was significantly higher than that at 20% FR, and no significant changes were observed at 40% and 50% FRs. Thus, 30% FR was recommended as the optimal carrier FR for the denitrifying MBBR.

## 1. Introduction

High total nitrogen (TN) of wastewater treatment plant (WWTP) effluent is always the primary factor affecting its compliance with the discharge standards and recycling, because the TN standard for WWTP effluent (Class I-A, 15 mg·L^−1^, SEPA of China, 2002) [1] is much higher than the maximum required TN value of natural surface water (Class V, 2.0 mg/L^−1^, SEPA of China, 2002) [2]. Therefore, it is very urgent to further remove the TN from WWTP effluent for the production of high-quality recycled water, which can be used as recharge water for rivers or groundwater in water shortage areas. The TN of WWTP effluent is primarily composed of ammonium, nitrite and nitrate, while nitrate accounts for more than 60% [3]. WWTP effluent always has low organic content and low carbon-to-nitrogen ratio (C/N), and it perhaps contains lead and zinc ions caused by the dissolution of minerals [4,5], which results in the complexity of WWTP effluent denitrification and nitrogen removal. It is difficult to achieve advanced nitrogen removal for WWTP effluent by traditional activated sludge processes. Some methods such as filtration and adsorption have been used for nutrient removal, which are often costly and difficult to maintain [6,7]. 

Moving bed biofilm reactors (MBBRs), which were developed in the 1980s for nitrogen removal, combine the advantages of activated sludge and biofilm in one reactor by dosing suspended carriers [8,9]. Biofilms, which attach and grow on the engineered carriers kept in constant suspension, are protected from abrasion when localized in the interior spaces of the MBBR carriers [10], and thus provide large surface area for the microbial colonization. Recent studies recommend MBBRs as promising alternative to the activated sludge systems with respect to the degradation of micropollutants [11,12,13,14]. MBBRs have been successfully employed to treat the municipal and industrial wastewater and upgrade small WWTPs [15,16]. Denitrifying MBBRs have been applied in the treatment of nitrate-contaminated wastewater as seawater, sewage, WWTP effluent, etc. [17,18,19,20], and good nitrate removal efficiency has been achieved. It has been reported that denitrifying MBBR filled with 30% polyethylene (PE) carrier obtained 61.9 ± 16.8% TN removal for the advanced WWPT effluent treatment, and the effluent TN concentration was less than 5 mg/L when the influent COD_added_/NO_3_^−^-N was kept at 4.6 by methanol addition as an external carbon source [21]. Liu et al. found that 25℃ was the optimum temperature for the removal of nitrogen from WWTP effluent by denitrifying MBBR, considering the nitrogen and organic removal efficiency as a whole [22]. Li et al. used raw *Arundo donax* pieces as carbon source and biofilm carrier in a lab-scale denitrifying MBBR to remove NO_3_^−^-N from reverse osmosis concentrate, with high denitrification capability (73.2% ± 19.5% NO_3_^−^-N removal efficiency and 8.10 ± 3.45 g NO_3_^−^-N/m^3^/d volumetric removal rate) obtained at stable operation stage [23]. 

In MBBR systems, the type and quantity of carriers directly affect the population distribution characteristics of microorganisms, and thus influence the wastewater treatment efficiency [20]. Quite some studies about MBBR carriers have been performed in recent years [24,25,26], and the carriers include granular activated carbon, sand, diatomaceous earth, polyethylene (PE), polyurethane foam (PUF) and several biodegradable materials [27,28]. PE carriers, which possess high porosity, are ideal microbial carriers for their microbial self-immobilization promotion, excellent mechanical strength and low-cost. Thus, they have been widely applied in various MBBRs, and achieved good pollutant removal efficiency for the treatment of different kinds of wastewater [29,30,31]. Four different carriers- PE, polypropylene (PP), PUF and haydite were investigated for their influence on the nitrogen removal efficiency of WWTP effluent treatment by denitrifying MBBR, and PE carriers were found to be the best type [20]. Moreover, some studies proved the feasibility and superiority of PE carriers in MBBR systems from various aspects [30,31].

There are many factors affecting the MBBR performance on sewage treatment, and the most critical one is the surface area available for biofilm growth, which is related to the mechanical characteristics and filling ratio (FR) of the carriers [32]. Gu et al. [33] demonstrated that the FR of PE carriers significantly affected the chemical oxygen demand (COD), phenol, isothiocyanate and total ammonium removal in coking wastewater treatment by MBBR systems. Barwal and Chaudhary [34] showed that the MBBR with 40% FR of PP carrier exhibited excellent performance for synthetic municipal wastewater treatment. The aerobic MBBR with 30% FR of sponge carrier achieved higher TN removal than those with 10% and 20% FRs [35]. Yuan et al. [36] reported that 96.3% ammonium removal efficiency was obtained by an aerobic MBBR with 40% FR of PUF carrier at 5 h hydraulic retention time (HRT), which was significantly higher than that achieved at 20% FR (37.4%). Additionally, 20% ~ 30% FR was recommended by Deng et al. [24] and Piculell et al. [37] in their MBBR systems. 

As mentioned above, quite some studies about the influence of FR on aerobic MBBRs have been carried out. However, the mechanisms of the FR effect on nitrogen removal and microbial characteristics for the denitrifying MBBR treatment of WWTP effluent have not been extensively investigated yet, and this study aimed to explore such mechanisms. A bench-scale denitrifying MBBR was constructed and operated at 20%, 30%, 40%, and 50% carrier FRs, the relationship between the contaminant removal and FR was explored, and the diversity of microbial communities in the biofilms was analyzed by quantitative polymerase chain reaction (qPCR) and terminal restriction fragment length polymorphism (T-RFLP) methods. Moreover, the optimal carrier FR for denitrifying MBBR was recommended. The objective of this study was to provide theoretical basis for the advanced treatment of WWTP effluent by denitrifying MBBR.

## 2. Materials and Methods 

### 2.1. Experimental Denitrifying MBBR and Carrier

A bench-scale denitrifying MBBR was assembled, which consisted of a plexiglass cylinder with 120 mm inner diameter, 500 mm height and 0.38 L tapered bottom (Figure 1). The total reactor volume was 6.03 L, and the effective volume was 5.65 L. The PE carriers were cylindrically shaped with 25 mm nominal diameter, 10 mm average length, 960–980 kg/m^3^ specific density and 620 m^2^/m^3^ specific surface area (Dalian Yudu Environmental Engineering Technology Co., Ltd., Dalian, China).

### 2.2. WWTP Effluent Characteristics and Experimental Design

The effluent from Beijing Yongfeng WWTP, which uses Carrousel 3000 oxidation ditch treatment process, was supplied as denitrifying MBBR influent with COD_added_/NO_3_^−^-N maintained at 6.7 by methanol addition. Yongfeng WWTP with 20,000 m^3^/d designed treatment capacity, which was built in December 2008 and affiliated to Beijing Bihai Environmental Technology Co., Ltd., is located in Haidian District, Beijing, China. The influent, whose characteristics are listed in Table 1, was continuously fed to the reactor by peristaltic pump (BT100-1L, Baoding Lange Constant Pump Company, Beijing, China). The reactor was inoculated by the activated sludge taken from the anoxic tank of Beijing Yongfeng WWTP, and the inoculation mixed liquid suspended solids (MLSS), mixed liquor volatile suspended solids (MLVSS), MLVSS/MLSS, settling velocity (SV), and sludge volume index (SVI) were 7000 mg/L, 3549 mg/L, 0.51, 66%, and 94 mL/g, respectively. At the beginning of the experiment, 2 L activated sludge was dosed with 4 L WWTP effluent in the reactor. Then, low flow rate was applied, which gradually increased to the scheduled HRT. The experiment was operated at four phases with different FRs, i.e., 20% (Phase I), 30% (Phase II), 40% (Phase III), and 50% (Phase IV). A heating rod was used to maintain the water temperature at 24–26 °C, while a propeller stirrer with 80 mm diameter and 30 rpm speed was used to mix the sludge, carrier and wastewater. The HRT of each phase was 12 h. The treatment performance was evaluated by the influent and effluent NO_3_^−^-N, NO_2_^−^-N, NH_4_^+^-N, TN, and COD analysis once every two days.

### 2.3. Sample Collection and Analysis

Water samples were collected from the inlet and outlet of the system during each operation phase and analyzed immediately at the Laboratory of Chinese Research Academy of Environmental Sciences. COD and NH_4_^+^-N were analyzed according to the standard methods [38]. TN was measured by TOC-V_CPH_ total organic carbon (TOC) analyzer (Shimadzu, Kyoto, Japan), while NO_2_^−^-N and NO_3_^−^-N were determined by ion chromatography (ICS-1000, DIONEX, California, USA). All samples for TN, NO_2_^−^-N, NO_3_^−^-N, and NH_4_^+^-N measurement were pretreated by 0.45 μm membrane filter.

### 2.4. Biofilm Characteristics

Biofilms on carriers of different phases at stable operation stage were compared using different techniques to characterize their physical features, microbial population dynamics and distribution.

#### 2.4.1. Biomass

The biomass of the carrier biofilm was determined as follows: A certain amount of carriers were taken from the reactor, submerged in 20 mL NaOH of 1 M in a clean tube, maintained at 80 °C for 30 min in water bath, ultrasonically treated for 1 min at 100 W to separate the fixed bacteria from the carrier surface, vortexed for 30 s to uniformly disperse the bacteria in solution [39], and then the dry weight of the biofilm was measured.

#### 2.4.2. Scanning Electron Microscopy (SEM)

The microbe distribution on the carrier surface was analyzed using SEM. An appropriate amount of biofilm-attached carriers were obtained from the denitrifying MBBR at stable stage. Approximately 5 × 5 mm biofilm-containing samples were cut from the obtained carriers, then fixed by 2.5% neutral glutaraldehyde and washed with phosphate buffer. Following ethanol gradient dehydration, the samples underwent critical point drying in CO_2_ critical point dryer (SPI Inc., PA, California, USA) and ion-sputtered with gold in Eiko Ion Coater (model IB-3, Hatachi Inc., Naka, Japan), and then their morphology was observed by SEM (HITACHI S-570 SEM, Hatachi Inc., Naka, Japan) at an accelerating voltage of 12 kV. 

#### 2.4.3. qPCR and T-RFLP Analysis

About 2 g wet biofilm abraded from the denitrifying MBBR carriers at stable operation stage was put into the sterile Eppendorf tube. Genomic DNA, which was extracted from the biofilm samples by UltraClean DNA Kit (Mobio Laboratories, California, USA) according to the protocol provided by the manufacturer, was detected by 1% agarose gel electrophoresis and stored at −20 °C for future use. *Nitrous oxide reductase* (*nosZ*) primers as *nosZ*-F (5’-CGYTGTTCMTCGACAGCCAG-3’ with the 5’-end labeled with carboxyfluorescein) and *nosZ*1622R (5’-CGSACCTTSTTGCCSTYGCG-3) [40] were used for DNA amplification. The PCR amplification conditions were as follows: 1 cycle at 94 °C for 5 min, 35 cycles at 95 °C for 0.5 min, 55 °C for 0.5 min and 72 °C for 1.5 min, final extension at 72 °C for 10 min [41]. Restriction enzyme (HhaI) was used to digest the PCR products purified with QIAquick PCR purification kit (Qiagen Inc, Berlin, Germany) at 37 °C for 3 h. The Shannon-Wiener index and Bray-Curtis similarity index, which are based on the concept of evenness or equitability, were calculated [42,43] to evaluate the bacterial species diversity and similarity among different samples. qPCR was conducted as follows: The *nitrate reductase* (*narG*) and *nosZ* genes were amplified by PCR, ligated into the pMD-19T vector, and transformed; the plasmid was extracted; the positive clones were identified with PCR, and then sequenced and identified (Beijing Nosy Genome Research Center, Ltd.).

## 3. Results and Discussion

### 3.1. Effects of FR on COD removal

At FRs of 20%, 30%, 40%, and 50%, the average COD removal rates were 33.8 ± 18.4%, 37.7 ± 17.1%, 47.0 ± 15.5%, and 55.2 ± 11.8%, respectively. The COD removal efficiency increased with the increase of FR, and all the effluent COD met the Class I(A) requirement of the Discharge Standard of Pollutants for Municipal WWTP in China (GB18918-2002) [1], i.e., less than 50 mg L^−1^. The above-mentioned effluent COD value was consistent with that of the denitrifying MBBR for WWPT effluent treatment [21].

### 3.2. Effects of FR on Nitrogen Removal

#### 3.2.1. Effects of FR on Ammonium, Nitrite Conversion and TN Removal

As shown in Table 2, the NH_4_^+^-N removal rates at 30% and 40% FRs were higher than those at 20% and 50% FRs. However, since low influent nitrogen load is more conducive to TN removal in biofilm systems [44], lower TN removal rates were observed at 30% and 40% FRs. Slight NO_2_^−^-N accumulation occurred in the reactor, which might have resulted from the activity inhibition of the nitrifying bacteria caused by limited dissolved oxygen. Another possible explanation for this phenomenon was that the microbe could not easily metabolize methanol, which thus led to insufficient available carbon in the system. Then, the carbon deficiency caused incomplete denitrification, in which nitrate was reduced to nitrite instead of nitrogen, and resulted in nitrite accumulation [45]. 

When appropriate FR was applied, the biofilms attached to the carriers would exist for long HRTs, which ensured the growth of autotrophic nitrifying microorganisms on the biofilms with long generation period and slow proliferation rate. The large gaps between the carriers promoted adequate contact among the solid, gas and liquid materials, which increased the mass transfer area and rate and then resulted in enhanced growth of the microorganisms [46]. As illustrated in Table 2, the turbulent conditions might be worsened when the FR is greater than 50%, and a large number of microorganisms might be stripped from the carrier surface due to the effects of mass transfer and hydraulic shearing, thus impeding the growth of the bacteria related with NH_4_^+^-N removal. Li et al. reported that the anammox gene existed in the carrier biofilm of denitrifying MBBR for reverse osmosis concentrate treatment [23]. The weakening of the anammox gene bacteria is bound to reduce the NH_4_^+^-N removal efficiency of the whole system. The NH_4_^+^-N removal efficiency trend observed in this study could be explained by the reasons mentioned above.

As a result, 30% and 40% FRs are recommended as the optimal FRs for denitrifying MBBR based on ammonium, nitrite conversion and TN removal.

#### 3.2.2. Effects of FR on Nitrate Removal

Figure 2 presents the NO_3_^−^-N removal capability at different FRs during the entire denitrifying MBBR operation period. The NO_3_^−^-N removal rates were 94.3 ± 3.9%, 87.7 ± 7.3%, 89.7 ± 11.6%, and 94.6 ± 4.0% at FRs of 20%, 30%, 40%, and 50%, while the effluent NO_3_^−^-N concentration was 0.2 ± 0.1 mg/L, 0.8 ± 0.5 mg/L, 0.6 ± 0.5 mg/L, and 0.2 ± 0.1 mg/L, respectively. The denitrification rate (r_NO3--N_) was obtained using linear regression of the NO_3_^−^-N concentration measured during the experiments. As shown in Figure 2, the denitrification rates at FRs of 20%, 30%, 40%, and 50% were 8.0 ± 5.6, 11.3 ± 4.6, 11.6 ± 4.6, and 10.0 ± 4.9 mgNO_3_^−^ -N/L/d, and higher denitrification rates were obtained at 30% and 40% FRs. The denitrification rate, which was closely related to the influent NO_3_^−^-N concentration, increased with the increase of influent NO_3_^−^-N [19].

Denitrifying bacteria are facultative heterotrophic microorganisms that grow rapidly and proliferate within a short period of time. In denitrifying MBBR systems, denitrifying bacteria need organic matter to act as electron donors for nitrate conversion. Therefore, the microorganism population increase in the system might lead to the decrease of microbial activity for the scarcity of carbon source. The denitrification rate increased with the increase of influent nitrate load, which might have been caused by the enhancement of denitrifying bacteria activity, and such results indicated that the system has great potential for the treatment of wastewater with high nitrate concentration. It is reported that a woodchip bioreactor with high NO_3_^−^-N load increased the denitrification rate in similar systems, which is consistent with the results of this study [47].

Therefore, FRs of 30% and 40% are recommended as the optimal ones for denitrifying MBBR in the aspect of nitrate removal.

### 3.3. Effects of FR on Microbial Community

#### 3.3.1. Effects of FR on Biomass

As shown in Table 2, the amount of microorganisms on carriers at 20% FR (4.86 mg/g carriers) was the largest, compared with those at 30%, 40%, and 50% FRs (3.29 mg/g carriers, 2.51 mg/g carriers, and 2.17 mg/g carriers, respectively), which might have resulted from the lower hydraulic shearing effect and the relatively stable growth environment for the microorganisms. However, the mass transfer between microorganisms and nutrients was hindered at 20% FR, which might lead to relatively low biofilm activity [48]. Moreover, lower total microorganism population was observed because of the smaller total surface area of the carriers available for microbial attachment. The ratio of the biofilm to the unit weight of carrier decreased with the increase of FR, but the total biofilm mass of the entire system increased correspondingly due to the greater number and larger support surface of the carriers [35]. The biofilm thickness on each carrier decreased with the increase of FR, which facilitated the mass exchange and biofilm renewal in the denitrifying MBBR and resulted in higher vitality [48] and denitrification rate of the microorganisms (Figure 2). However, if the FR was too high (greater than 50%), the anoxic zone on the carrier surface would reduce due to the thinner biofilm, which would increase the aerobic microorganisms and then cause its carbon competition with the denitrifying bacteria. Aerobic microorganisms consume large amounts of organic matters in the water, which decreases the denitrification rate of the denitrifying bacteria for insufficient electron donors [49]. The theories mentioned above could explain the increase of COD removal rate and the decrease of denitrification rate at high FR conditions in this study.

#### 3.3.2. Effects of FR on Biofilm Thickness and Appearance

In general, the biofilms attach to the carriers in two different forms, i.e., thick and dense biofilm developed on the carrier surface, and biofilm deposited or entrapped in the interior voids of the carriers [36]. It can be seen from the SEM micrographs of the PE carrier obtained after acclimatization that thick and dense biofilms had formed. As shown in Figure 3, the biofilms were primarily composed of cocci, bacillus, filamentous bacteria and extracellular polymeric substances (EPS) under all experimental FR conditions, while cocci and bacillus accounted for the majority. Bacilli, cocci, and EPS assembled together, formed z*oogloea* and adhered to the carrier surface. At 20% FR, quite some viscous substances, cocci and bacillus presented together on the biofilms, formed compact microbial structure. The filamentous bacteria population increased and the biofilm thickness decreased with the increase of FR, which led to looser microbial structure, and similar results were observed in other studies [48]. Moreover, SEM was used to determine the biofilm composition on the carrier surface, which discovered that the biofilms in the denitrification system were primarily composed of rod bacteria and cocci [20], and rod bacteria were the predominant species in the denitrifying packed bed bioreactors [50]. The bacteria morphology of this study is similar with that of the reports mentioned above.

#### 3.3.3. Effects of FR on Microbial Abundance

Denitrification requires the synergy of multiple microorganisms and enzymes (Figure 4). *nosZ* and *narG* are the key genes needed for the conversion of nitrous oxide to nitrogen during the microbial denitrification process [51], and the abundance variation of these two genes can reflect the denitrification capacity [52]. In this study, qPCR based on *nosZ* and *narG* genes were used to evaluate the nitrogen removal capacity, which is presented in Table 3. Both the *nosZ* and *narG* gene abundance, which were increased with the increase of FR, were remarkably lower at 20% FR than those at other FRs, and they were within an order of magnitude at FRs of 30%, 40% and 50%. These results demonstrated that the increase of FR could not significantly improve the abundance of denitrifying bacteria genes when the FR was greater than 30%, which was consistent with the variation trends of the denitrification rate. 

#### 3.3.4. Effects of FR on Microbial Community Diversity

T-RFLP, which is a culture-independent molecular genetic method for generating profiles or fingerprints of environmental microbial communities [53,54], is a common tool in the characterization of microbes. In this study, *nosZ*-based T-RFLP was used to examine the influence of FR on the community diversity of denitrifying microbes. HhaI digestion of the PCR products illustrated the distribution of the resulting terminal restriction fragment (TRF) peaks, whose numbers and relative abundance varied among the profiles, and the results indicated that the sampled bacterial communities differed in their diversity and species composition (Figure 5). The predominant fragment observed at 20%, 30%, and 40% FRs was 168 bp (HhaI), which accounted for 84.7%, 32.4%, and 47.8%, respectively. The predominant fragment observed at 50% FR was of 31.5% relative abundance with 250 bp. The 168 bp and 250 bp fragments might represent the primary denitrifying bacteria in the denitrifying MBBR. The Shannon-Wiener index (H′) and evenness (E′) at different FRs are presented in Table 3. The functional diversity of the biofilm microbial community at 30% FR was higher than that at 20%, 40%, and 50% FRs. At 30% FR, the evenness, which increased with the increase of FR, was significantly higher than that at 20% FR. However, the evenness variation was not obvious when the FR was greater than 30%, and the highest evenness was achieved at 50% FR, which was consistent with other reports about the MBBR systems [9]. 

### 3.4. Analysis about the Application to Other Scale Reactors

Overall, 30% and 40% are recommended as the optimal FRs for denitrifying MBBR based on the ammonium and nitrite conversion, TN and nitrate removal. The microbial community diversity, evenness and denitrifying bacteria functional gene abundance (*nosZ* and *narG*) of the biofilms at 30% FR were significantly higher than those at 20% FR, while no significant differences were observed at 30%, 40%, and 50% FRs. Therefore, 30% is recommended as the optimum FR for denitrifying MBBR, which provides theoretical support for the construction and operation of the pilot-scale and full-scale denitrifying MBBR for the treatment of WWTP effluent as recharge water for rivers or groundwater.

## 4. Conclusions

The FR influence of the PE carrier on the performance and microbial characteristics of denitrifying MBBR for the treatment of WWTP effluent was extensively investigated by a bench-scale reactor at 12 h HRT, 6.7 COD_added_/NO_3_^−^-N and 24–26 °C temperature, and the conclusions were as follows:

(1) Excellent performance in terms of COD and nitrogen removal was achieved using PE carriers at FRs of 20%, 30%, 40%, and 50%. The NO_3_^−^ -N removal efficiency and denitrification rates at 20%, 30%, 40%, and 50% FRs were 94.3 ± 3.9% and 8.0 ± 5.6 mg NO_3_^−^ -N/L/d, 87.7 ± 7.3% and 8.0 ± 5.6 mg NO_3_^−^ -N/L/d, 89.7 ± 11.6% and 8.0 ± 5.6 mg NO_3_^−^ -N/L/d, and 94.6 ± 4.0% and 10.0 ± 4.9 mg NO_3_^−^ -N/L/d, respectively. No remarkable changes were observed for NO_3_^−^ -N removal efficiency at different FRs, but higher denitrification rates were obtained at 30% and 40% FRs.

(2) The biomass of the carrier biofilm decreased with the increase of FR, but the total biofilm biomass of the entire system increased due to the greater number of carriers. The SEM results demonstrated that the biofilms were primarily composed of cocci, bacillus, filamentous bacteria and EPS at all FRs, while cocci and bacillus accounted for the majority. The filamentous bacteria population increased and the biofilm thickness decreased with the increase of FR. 

(3) The microbial community diversity, evenness, and denitrifying bacteria functional gene abundance (*nosZ* and *narG*) of the biofilms at 30% FR were significantly higher than those at 20% FR, and no significant differences were observed at 30%, 40%, and 50% FRs. 

(4) Overall, 30% is recommended as the optimum FR for denitrifying MBBR, which provides theoretical support for the construction and operation of the pilot-scale and full-scale denitrifying MBBR for the treatment of WWTP effluent as recharge water for rivers or groundwater.

## Figures and Tables

**Figure 1 ijerph-16-03244-f001:**
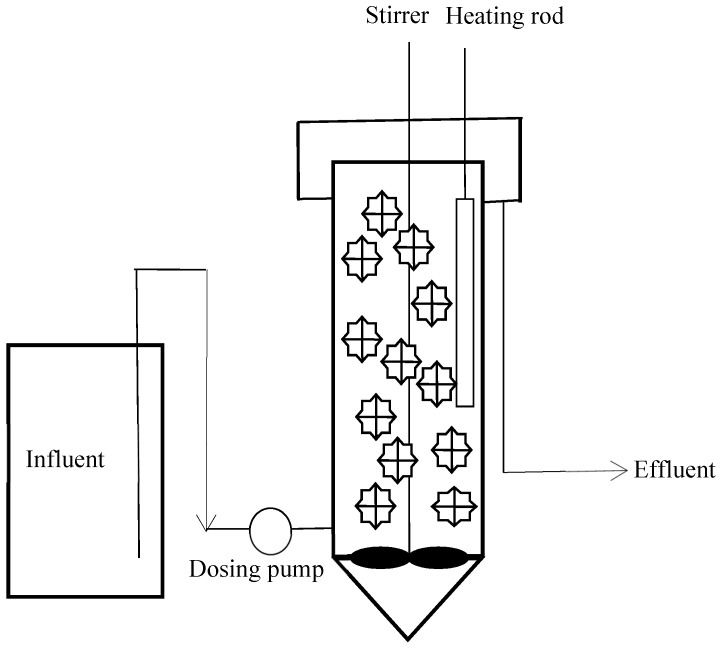
Schematic diagram of the laboratory denitrifying MBBR set-up.

**Figure 2 ijerph-16-03244-f002:**
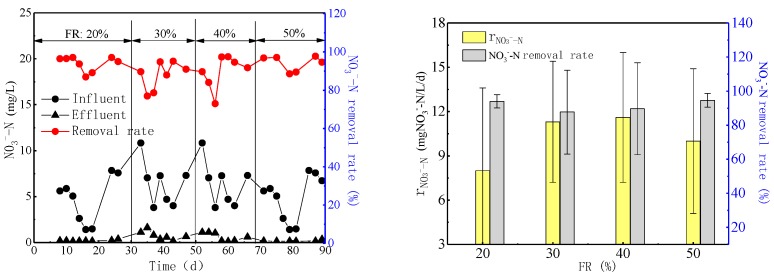
NO_3_^−^-N removal rate and denitrification rate (r_NO3--N_) at different FRs.

**Figure 3 ijerph-16-03244-f003:**
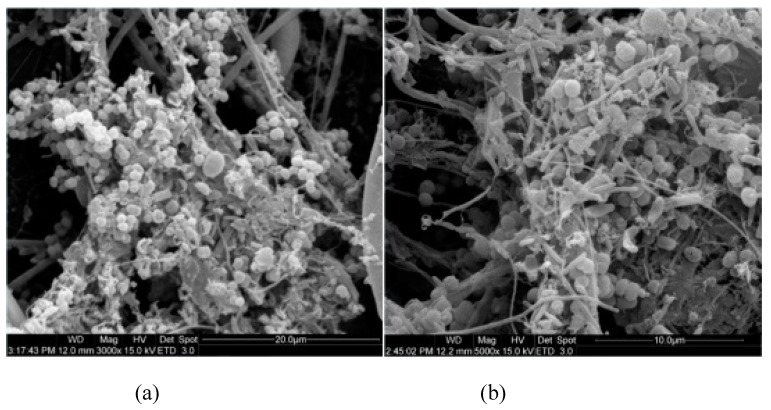
SEM micrographs of the biofilm at different FRs: (**a**) 20%, (**b**) 30%, (**c**) 40% and (**d**) 50%.

**Figure 4 ijerph-16-03244-f004:**
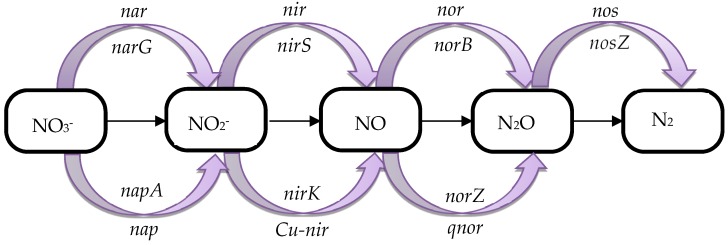
*nosZ* community structure at different FRs.

**Figure 5 ijerph-16-03244-f005:**
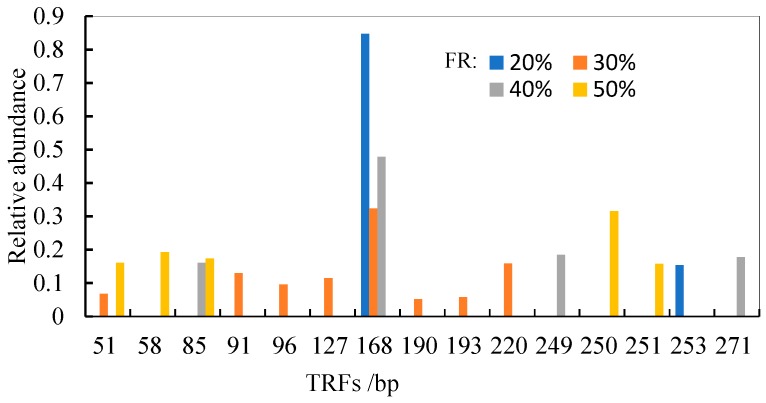
*nosZ*-based community structure at different FRs.

**Table 1 ijerph-16-03244-t001:** Water quality of the WWTP effluent used as influent.

Phase	Operational Day (d)	pH	NO_3_-N (mg/L)	NO_2_-N (mg/L)	NH_4_^+^-N (mg/L)	TN (mg/L)	COD * (mg/L)
I	0–30	7.2–7.9	4.7 ± 2.6	0.6 ± 0.2	0.7 ± 0.2	8.7 ± 5.2	52.2 ± 25.3
II	31–50	7.1–7.8	6.4 ± 2.5	0.8 ± 0.4	1.4 ± 0.5	11.0 ± 3.0	68.9 ± 9.0
III	51–70	7.1–7.8	6.4 ± 2.5	0.8 ± 0.4	1.4 ± 0.5	11.0 ± 3.0	68.9 ± 9.0
IV	71–90	7.2–7.9	5.1 ± 2.4	0.7 ± 0.2	0.9 ± 0.6	8.9 ± 4.8	52.2 ± 25.3

* COD is the corresponding value of 6.7 COD_added_/NO_3_^−-^N achieved by methanol addition.

**Table 2 ijerph-16-03244-t002:** NH_4_^+^-N, NO_2_^−^-N, and TN removal efficiency and biofilm mass of denitrifying MBBR at different FRs.

FR(%)	NH_4_^+^-N (mg/L)			NO_2_-N (mg/L)			TN (mg/L)			Biofilm mass (mg/g-carriers)
	Influent	Effluent	Removal rate (%)	Influent	Effluent	Removal rate (%)	Influent	Effluent	Removal rate (%)	
20	0.7 ± 0.2	0.5 ± 0.3	27.9 ± 39.1	0.6 ± 0.2	0.8 ± 0.5	−21.1 ± 61.9	8.7 ± 5.2	2.6 ± 3.2	74.5 ± 16.0	4.86
30	1.4 ± 0.5	0.8 ± 0.4	38.5 ± 28.0	0.8 ± 0.4	0.9 ± 0.4	−29.6 ± 65.7	11.0 ± 3.0	6.2 ± 2.2	42.1 ± 16.2	3.29
40	1.4 ± 0.5	0.9 ± 0.5	35.0 ± 31.9	0.8 ± 0.4	1.0 ± 0.6	−27.3 ± 54.6	11.0 ± 3.0	6.2 ± 2.6	42.8 ± 20.0	2.51
50	0.9 ± 0.6	0.6 ± 0.2	19.4 ± 39.6	0.7 ± 0.2	0.7 ± 0.3	−4.9 ± 26.7	8.9 ± 4.8	1.9 ± 2.0	80.4 ± 11.4	2.17

**Table 3 ijerph-16-03244-t003:** Distribution of denitrifying bacteria functional genes and Shannon-Wiener index with evenness at different FRs.

Filling rate	20%	30%	40%	50%
*narG* abundance (copies/g-SS)	2.76 × 10^7^	1.00 × 10^8^	2.78 × 10^8^	2.89 × 10^8^
*nosZ* abundance (copies/g-SS)	7.66 × 10^4^	1.74 × 10^7^	2.15 × 10^7^	2.42 × 10^7^
Shannon-Wiener index (H′)	0.4	1.9	1.3	1.6
Evenness (E′)	0.6	0.9	0.9	1.0

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
