# Peer review of "Influence of Carrier Filling Ratio on the Advanced Nitrogen Removal from Wastewater Treatment Plant Effluent by Denitrifying MBBR"

_ijerph, 2019, doi:10.3390/ijerph16183244_

Round 1

Reviewer 1 Report

The manuscript deals with the conditions of wastewater treatment by MBBR technique. The authors used a large number of good-quality references to support the literature review and the discussion section. The work may be important for the degree of detail and characterization with which this topic has been treated. However, some points in the manuscript require revision by the authors.

Dear authors,

my suggestions are shown below:

1. Abstract: please re-arrange this part in accordance with the Instructions for Authors (underlined): “1) Background: Place the question addressed in a broad context and highlight the purpose of the study; 2) Methods: Describe briefly the main methods or treatments applied; 3) Results: Summarize the article's main findings; and 4) Conclusion: Indicate the main conclusions or interpretations).”

2.  Introduction: The authors should better emphasize the novelty of their work compared to previous (cited) studies, its significance and highlight the main conclusions (Instructions for Authors).

3. References: In the text, reference numbers should be placed in square brackets – please delete the names of the authors. All references should be written in accordance with the Instructions for Authors.

4. Please complete Author Contributions and Conflicts of Interest.

5. Conclusions: Please explain to what extent the results may be useful for the operation of WWTP. I also suggest giving more information on the impact of FR on microorganisms – shortcut of the conclusions from the chapters 3.2.1-3.3.4.

6. No information is reported on the characteristic of the Beijing Yongfeng WWTP -  please give some details.

7. The information was duplicated (chapter 3.1 and fig. 2). I suggest to remove fig.2.

8. I suggest to combine tables 2 and 3 and also table 4 with table 5.

Specific comments - see attached file.

Reviewer 2 Report

In this work, the carrier FR used in the denitrification MBBR treatment of secondary sedimentation tank effluent at 12 h HRT was optimized, and excellent performance in terms of COD and nitrogen removal was achieved. This topic is interesting and the study is significant. However, it needs minor revision before it is published in this journal. The following issues should be carefully addressed.

1. Some language problem should be further revised in the manuscript.

2. The authors indicated that this study is to provide theoretical support for the advanced treatment of WWTP effluent by denitrification MBBR. Actually, wastewater contained lead and zinc ions caused by various contamination, such as the dissolution of minerals, which will result in the complexity for wastewater denitrification and greatly affect the nitrogen removal. Thus, this phenomenon should be added in the “Introduction”, and the relevant references should be added (Surface modification of smithsonite with ammonia to enhance the formation of sulfidization products and its response to flotation, Minerals Engineering 137 (2019) 1–9; Combined DFT and XPS investigation of enhanced adsorption of sulfide species onto cerussite by surface modification with chloride, Applied Surface Science, 2017, 425: 8–15).

3. The samples and testing condition description should be added in “2.4.2. Scanning electron microscopy (SEM)”.

4. Give more detail analysis and discussion in the section of “Discussion”, and it should be amended by including a brief analysis respect to the use of the reported results to other scale.

5. Conclusions should be rearranged.

Reviewer 3 Report

I could not find the novelty of your paper. But your obtained results will contribute the development of denitrifying MBBR. The level of your paper did not reach the acceptance level of this journal. I have judged the major revision is required before accepting your paper.

1.     English is poor. Native check is required.

2.     There are lot of sentence with consecutive noun. More than series of three nouns is not allowed in the paper.

3.     Line 17  denitrification moving bed → denitrifying moving bed

4.     Line 21 at 20%, 30% , 40% and 50% FRs → with FRs of 20%, 30% , 40% and 50%

5.     Line 22  mg NO3-N(L-1d-1) → mgNO3-N/L/d

6.     You must use ammonium instead of ammonia.

7.     You must write the enzyme name in italic form.

8.     Line 51  61.9±16.8 TN → 61.9±16.8 of TN

9.     Line 51  effluent T-N is → effluent T-N concentration 

10.  Line 63  prove the feasibility  → proved the feasibility

11.  Line 79  An experimental denitrification MBBR → Bench scale denitrifying MBBR

12.  Mixing is very important for your bench-scale reactor. You must write clearly the mixing system (size and shape of impeller and mixing speed) in your paper.

13.  You must write the seeding source and acclimation of way for denitrifying MBBR. Especially, the acclimation is essential for the establishment of denitrifying reactor using methanol as H-donor.

14.  Title of Table 1  → Water quality of WWTP effluent used as influent

15.  Fig.2 and 3   horizontal axis  → please remove % and show filling ratio(%)

16.  Line 222  How did you distinguish the Zoogloea bacteria in SEM photos?

17.  Fig. 5  You must missed the expression of nitrogen compounds.

(NO → NO2-, N2 → N2O)

18.  Reference (17)  → You must show the information on volume and page.

Round 2

Reviewer 1 Report

Thank you for the manuscript revision. I have no more suggestions.

Reviewer 3 Report

You have revised your paper by taking into accounts my comments. But your English is still poor. I strongly request to take native check.
